# Endo-4DTS: Monocular 4D Scene Synthesis for Endoscopy via Deformable Triangle Splatting

**Laura Salort-Benejam** (ORCID)                          LAURA.SALORT@UPC.EDU
**Antonio Agudo** (ORCID)                                 ANTONIO.AGUDO@UPC.EDU
*Institut de Robòtica i Informàtica Industrial, CSIC-UPC, Barcelona, Spain*

**Editors:** Accepted for publication at MIDL 2026

## Abstract

Endoscopy is an essential procedure in medical imaging, routinely applied for diagnostic, prognostic and therapeutic purposes. Developing robust methods for 3D reconstruction of endoscopic videos has the potential to improve the visualization of complex anatomies, increase diagnostic accuracy, and guide surgical procedures. Despite recent advancements the task remains highly challenging. The deformable nature of soft tissues makes classical computer-vision algorithms useless, and additional difficulties arise from the widespread use of monocular cameras, unknown camera parameters, occlusions, illumination changes, motion blur and other artifacts. In this work, we present Endo-4DTS, a novel self-supervised pipeline based on triangle splatting for 4D scene synthesis of deformable endoscopy scenes from monocular videos with a static camera, the first time this type of solution is proposed to endoscopic images and in time-varying tissues. Our approach represents the endoscopic environment with a canonical set of triangles, optimized jointly with a deformation network, enabling consistent 4D synthesis of dynamic tissues. We incorporate additional geometric and depth-based objectives that further guide learning in the challenging context of deformable endoscopic scenes. Experiments on several endoscopic videos with non-rigid tissues, occlusions and illumination changes, show that Endo-4DTS reliably captures soft-tissue deformations in endoscopic scenes. We demonstrate that Endo-4DTS consistently outperforms previous state-of-the-art methods across multiple metrics.

**Keywords:** Triangle splatting, differentiable rendering, endoscopy, non-rigid tissues.

## 1. Introduction

Endoscopy has become an essential medical imaging modality for examining the human body across a wide range of interventions, for diagnostic, prognostic and therapeutic purposes. Despite the existence of stereo and RGB-D cameras that provide additional depth information, their larger size requires bigger incisions and limits clinical applicability. As a result, monocular cameras, with their compact design and versatility, remain the most widely used in endoscopic devices today (Edwards et al., 2022; Boese et al., 2022). The wide variety of possible endoscopic interventions–such as colonoscopies, bronchoscopies or arthroscopies– presents a major challenge for recovering 3D information from visual cues, as the models must be able to adapt to significant changes in anatomy, appearance, illumination, camera motion and tissue deformation. As a result, there is a strong need for generic, robust models capable of interpreting visual endoscopic data across different anatomies and deformation patterns. These would benefit both patients and clinicians by enabling 3D visualization of anatomical structures, facilitating the assessment of regions that are difficult

to inspect due to restricted viewpoints, and allowing patient-specific reconstructions to be revisited during follow-up examinations to monitor disease progression.

However, recovering 3D information from monocular videos is fundamentally challenging. Classical methods–including rigid and non-rigid Structure-from-Motion (SfM) (Agarwal et al., 2009; Schönberger and Frahm, 2016; Agudo et al., 2016; Agudo, 2020; Gómez-Rodríguez et al., 2022), shape-from-template (Lamarca et al., 2021), shape-from-shading (Rodríguez-Puigvert et al., 2023), photometric stereo (Collins and Bartoli, 2012), and supervised approaches (Rau et al., 2023)–typically depend on explicit correspondences or strong priors on motion and deformation to solve the problem. These assumptions are rarely satisfied in endoscopies, where biological tissues suffer non-rigid deformations, making 3D reconstruction inherently ill-posed. Moreover, endoscopic videos introduce additional challenges that can decrease the robustness of the shape and camera estimations of traditional algorithms, such as specular highlights caused by the light source, abrupt camera movements, limited and highly constrained viewpoints, occlusions from tools or fluid, low-texture regions, motion blur, illumination changes, etc.

Recent advances in neural rendering have greatly improved 3D reconstruction under challenging imaging conditions. Neural Radiance Fields (NeRF) (Mildenhall et al., 2020) introduced a powerful implicit representation that enables high-quality novel-view synthesis through differentiable volume rendering. Despite numerous extensions for pose estimation (Jeong et al., 2021; Wen et al., 2023), acceleration (Müller et al., 2022; Fridovich-Keil et al., 2022), and dynamic scenes (Pumarola et al., 2021; Park et al., 2021; de Paco and Agudo, 2024), NeRF remains limited by its reliance on accurate camera poses, heavy computation times, and difficulty handling uncontrolled lighting and non-rigid motion. Early adaptations to endoscopy settings such as EndoNeRF (Wang et al., 2022) and follow-up variants (Zha et al., 2023; Yang et al., 2023a, 2024a; Salort-Benejam and Agudo, 2026) assume fixed cameras and depend on stereo or auxiliary depth cues to solve the 3D reconstruction problem on deformable scenes, limiting their applicability to many endoscopy procedures where such information is not available.

The recent emergence of 3D Gaussian Splatting (3DGS) (Kerbl et al., 2023) offers a more efficient explicit representation of the scene through a set of learnable anisotropic 3D Gaussians, enabling real-time rendering with significantly faster optimization. However, its formulation assumes rigid scenes and requires SfM camera calibration and point-cloud for initialization, which would be unfeasible in endoscopy settings as this classical method fails when faced with non-rigid scenes. Several works have extended the original work for deformable environments (Luiten et al., 2024; Li et al., 2024; Yang et al., 2023b; Lin et al., 2024; Zhu et al., 2024b; Bae et al., 2024), typically by using a neural network to model deformations. Others have adapted them to endoscopy (Liu et al., 2024; Huang et al., 2024; Zhu et al., 2024a; Xie et al., 2024; Yang et al., 2024b), however they still rely on rigid-camera assumptions, SfM, or external depth estimation. Meanwhile, works on camera estimation for endoscopy (Bonilla et al., 2024; Wang et al., 2024) do not explicitly model tissue deformation.

One of the latest works on 3D scene reconstruction and differentiable rendering is triangle splatting (Held et al., 2026), that proposes using triangles as primitives for efficient high-quality scene representation. By using a set of unstructured disconnected triangles this approach leverages the latest advancements in computer graphics for GPU-accelerated

triangle processing and incorporates it in a fully-differentiable pipeline. This approach has shown remarkable results in visual fidelity, training and rendering speed as well as high-quality novel view synthesis, but it is only intended for rigid scenes with known camera parameters, usually calibrated with SfM, and also requires the resulting sparse point cloud for initialization.

In this work we propose Endo-4DTS to extend the *triangle splatting* approach to deformable scenes, specially for endoscopic procedures, without relying on additional priors such as known camera calibration, pre-computed templates or feature-based approaches for initialization. To the best of our knowledge, this work is the first to adapt the triangle splatting-based approach to deformable scenes, and the first to apply it to endoscopies.

## 2. Method

Our work builds on triangle splatting (Held et al., 2026), which represents a 3D scene using learnable triangle primitives, similar to 3DGS (Kerbl et al., 2023) but replacing the Gaussians with triangles. Each triangle is defined by three vertices $\mathbf{v}_i \in \mathbb{R}^3$, a color $\mathbf{c}$ represented with Spherical Harmonics, a smoothness parameter $\sigma$, and an opacity $o$. Triangles are initialized from a sparse point cloud obtained via SfM and refined through adaptive densification and pruning strategy as in (Kheradmand et al., 2024).

Triangle splatting (Held et al., 2026) first projects the 3D triangles to the image plane and then computes the final pixel values with point-based rendering, making it less computationally demanding than NeRF-based approaches. This projection of the 3D triangles to image space is done using a standard pinhole camera model $\overline{\mathbf{p}}_i = \mathbf{K}(\mathbf{R}_{cam}\mathbf{v}_i + \mathbf{t}_{cam})$, where $\overline{\mathbf{p}}_i \in \mathbb{R}^3$ is the homogeneous coordinate in image space of the projected triangle vertex $\mathbf{v}_i \in \mathbb{R}^3$, $\mathbf{p}_i \in \mathbb{R}^2$ is the same pixel coordinate in Euclidean space, $\mathbf{K}$ is a 3×3 known intrinsic camera matrix, and $\mathbf{R}_{cam}$ and $\mathbf{t}_{cam}$ are the rotation and translation, respectively, that define the camera pose. The projected triangles are then sorted by distance to the camera and the color $\zeta$ for each pixel $\mathbf{p}$ is obtained by accumulating the contribution of each overlapping triangle, as defined by the rendering equation in previous works (Kerbl et al., 2023; Held et al., 2025):

$$\zeta(\mathbf{p}) = \sum_{n=1}^{N_T} \mathbf{c}_n o_n \mathcal{W}(\mathbf{p}) \prod_{j=1}^{n-1} \big(1 - o_j \mathcal{W}(\mathbf{p})\big), \tag{1}$$

where $N_T$ is the total number of triangles, $\mathbf{c}_n$ is the learned color of the $n$-th triangle, and $\mathcal{W}(\mathbf{p})$ is a window function that smoothly influences the contribution of each projected triangle based on the distance from the triangle's incenter $\mathbf{s}_{inc} \in \mathbb{R}^2$, that is defined as:

$$\mathcal{W}(\mathbf{p}) = \text{ReLu}\left(\frac{\rho(\mathbf{p})}{\rho(\mathbf{s}_{inc})}\right)^{\sigma}. \tag{2}$$

Here, the learnable parameter $\sigma > 0$ controls the sharpness of the window function over $\rho$–the Signed Distance Field (SDF) of the triangle in image space. Smaller values of $\sigma$ produce sharper boundary transitions while larger values result in smoother transitions from the triangle boundary towards the incenter $\mathbf{s}_{inc}$, where $\mathcal{W}(\mathbf{p})$ attains its maximum value. The SDF is expressed as:

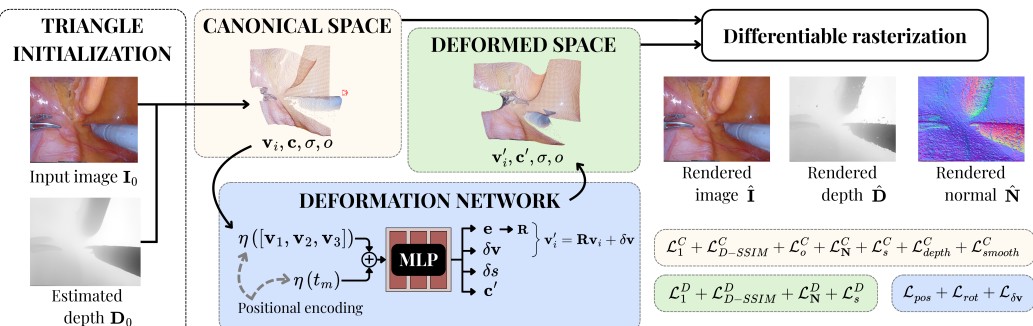

Figure 1: **Overview of our Endo-4DTS** pipeline for deformable triangle splatting from monocular endoscopy videos with static camera. Our method decomposes the structure information in canonical and deformed spaces, to capture the rigid and non-rigid contributions, respectively.

$$\rho(\mathbf{p}) = \max_{i \in \{1,2,3\}} \mathbf{n}_i \cdot \mathbf{p} + d_i^{SDF}, \tag{3}$$

where $\mathbf{n}_i$ are defined as the unit normals of the triangle edges pointing outwards and $d_i$ are the offsets making the triangle the zero-level set of $\rho$.

In this work, we propose Endo-4DTS, a framework that extends triangle splatting (Held et al., 2026) for 3D estimation of deformable scenes. Our approach introduces a canonical space representation in which a static set of 3D triangles is jointly optimized with a deformation network that estimates the rotation, translation, scale and color changes of these canonical triangles to the deformed space, see the proposed pipeline in Figure 1. Several additional losses are used to improve the photometric and geometric properties of the triangle representation.

Given a monocular input video containing deformable tissues, our model takes as input $\left\{ (\mathbf{I}_m, \mathbf{D}_m, t_m)_{m=1}^M, \mathbf{P}, f \right\}$, where $M$ is the total number of frames. Specifically, $\mathbf{I}_m \in \mathbb{R}^{H \times W \times 3}$ represents the $m$-th monocular RGB video frame, with height H and width W, $\mathbf{D}_m \in \mathbb{R}^{H \times W}$ is the corresponding estimated monocular depth map, $t_m = m/M \in [0,1]$ denotes the normalized time index of the $m$-th frame, $\mathbf{P} \in \mathbb{R}^{4 \times 4}$ represents the static camera pose, and $f$ is the focal length of the camera used to compose the intrinsic camera matrix $\mathbf{K}$. While our model assumes a fixed camera pose across time, this assumption is reasonable for many endoscopic procedures in which the endoscope is intentionally kept static to maintain a stable field of view to allow precise tissue manipulation by the physician. It is worth noting that camera motion may occur in gastroscopies and exploratory procedures, however our formulation could easily incorporate per-frame extrinsic information, such as $\mathbf{P}_m \in \mathbb{R}^{4 \times 4}$, when available.

To make our method applicable to a wider set of videos, we avoid the need for stereo-depth inputs by using video depth anything (Chen et al., 2025b), a pre-trained monocular depth estimation algorithm specially tailored for time consistency in videos, as a pseudo ground truth for our depth regularization.

## 2.1. Deformation network

The proposed deformation network $G_\Phi$ consists of an 8-layer Multilayer Perceptron (MLP) that takes as input the position of the triangles in the canonical space $\mathbf{v}_i$ and the time $t_m$ of the current frame. Following (Mildenhall et al., 2020), separate positional encoding $\eta$ of the inputs is applied to avoid overly smoothed representations, encouraging the MLP to learn high-frequency functions to represent the scene.

The deformation network will then predict for each triangle a rotation $\mathbf{e} = (\theta, \psi, \phi)$, encoded using Euler angles, a translation $\delta\mathbf{v} \in \mathbb{R}^3$, a scaling offset $\delta s \in \mathbb{R}$, and a new RGB color $\mathbf{c}' \in \mathbb{R}^3$:

$$\big(\mathbf{e}, \delta\mathbf{v}, \delta s, \mathbf{c}'\big) = G_\Phi\Big(\eta\big([\mathbf{v}_1, \mathbf{v}_2, \mathbf{v}_3]\big), \eta(t)\Big). \tag{4}$$

Following the pitch-roll-yaw convention we obtain the final rotation matrix $\mathbf{R} \in SO(3)$ and define the homogeneous transform $\mathbf{Q} = \begin{bmatrix} \mathbf{R} & \delta\mathbf{v} \\ \mathbf{0} & 1 \end{bmatrix} \in SE(3)$ and apply it to the triangles in the canonical space to obtain their transformed position as $\mathbf{v}_i' = \mathbf{Q}[\mathbf{v}_i^\top \ 1]^\top = \mathbf{R}\mathbf{v}_i + \delta\mathbf{v}$.

## 2.2. Optimization

The final loss consists of several terms, those used to optimize the canonical scene, denoted by the superscript $^C$, and those applied to the deformation network, indicated by the superscript $^D$:

$$\begin{aligned}\mathcal{L} = &(1 - \lambda_1)\mathcal{L}_1^C + \lambda_1\mathcal{L}_{D-SSIM}^C + \lambda_2\mathcal{L}_o^C + \lambda_3\mathcal{L}_\mathbf{N}^C + \lambda_4\mathcal{L}_s^C + \lambda_5\mathcal{L}_{depth}^C + \lambda_6\mathcal{L}_{smooth}^C \\ &+ (1 - \lambda_7)\mathcal{L}_1^D + \lambda_7\mathcal{L}_{D-SSIM}^D + \lambda_8\mathcal{L}_\mathbf{N}^D + \lambda_9\mathcal{L}_s^D + \lambda_{10}\mathcal{L}_{pos}^D + \lambda_{11}\mathcal{L}_{rot}^D + \lambda_{12}\mathcal{L}_{\delta\mathbf{v}}^D,\end{aligned} \tag{5}$$

where $\lambda_{1-12}$ are weighting factors. Next, we define every term in the global loss.

**Color loss:** Photometric fidelity is enforced using $\mathcal{L}_1$, the $l_1$-norm loss between the input and rendered images and a Differential-Structural Similarity Index Measure (SSIM) loss $\mathcal{L}_{D-SSIM} = 1 - SSIM(\mathbf{x}, \mathbf{y})$, in both canonical and deformed spaces.

**Opacity loss $\mathcal{L}_o^C$:** Opacity is regularized to avoid overly transparent or saturated triangles in the canonical space following (Kheradmand et al., 2024) $\mathcal{L}_o^C = \frac{1}{N_T} \sum_{n=1}^{N_T} |o_n|$, where $o_n$ is the opacity of the $n$-th triangle and $N_T$ the total number of them.

**Size loss $\mathcal{L}_s^C$ and $\mathcal{L}_s^D$:** Small or degenerate triangles are penalized in the canonical and transformed spaces via size regularization $\mathcal{L}_s = 2\|(\mathbf{v}_1 - \mathbf{v}_0) \times (\mathbf{v}_2 - \mathbf{v}_0)\|_2^{-1}$.

**Depth loss $\mathcal{L}_{depth}^C$:** We incorporate a depth loss term to the canonical representation to encourage the triangle soup to maintain triangles close to the pseudo-ground truth surface, defined as the $l_1$-norm loss between the estimated and input depth maps as:

$$\mathcal{L}_{depth}^C = \left\| \hat{\mathbf{D}}(\mathbf{p}) - \mathbf{D}(\mathbf{p}) \right\|_1. \tag{6}$$

**Depth smoothness loss $\mathcal{L}_{smooth}^C$:** Additionally, we incorporate a depth smoothness loss, similar to (Wang et al., 2018), enforcing neighboring pixels to have close depth values by using second-order gradients of the estimated depth:

$$\mathcal{L}_{smooth}^C = e^{-\nabla^2 \mathbf{D}(\mathbf{p})}\Big(\big|\nabla_{xx}\hat{\mathbf{D}}(\mathbf{p})\big| + \big|\nabla_{xy}\hat{\mathbf{D}}(\mathbf{p})\big| + \big|\nabla_{yy}\hat{\mathbf{D}}(\mathbf{p})\big|\Big), \tag{7}$$

where $\nabla^2 \mathbf{D}(\mathbf{p})$ is the Laplacian of the input depth, whose exponential is used as a weighting term to assign less importance to pixels that are more likely to be edges and discontinuities.

**Normal loss $\mathcal{L}_{\mathbf{N}}^C$ and $\mathcal{L}_{\mathbf{N}}^D$:** This term is applied to both the canonical and deformed spaces to encourage the rendered triangles normals $\mathbf{n}(\mathbf{p})$ to align with the pseudo-ground truth surface normals $\mathbf{N}(\mathbf{p})$ using:

$$\mathcal{L}_{\mathbf{N}}^C = 1 - \mathbf{n}(\mathbf{p})^\top \mathbf{N}(\mathbf{p}). \tag{8}$$

**Rotation and translation losses:** These two terms are used to encourage the deformation network to estimate rotations and translations close to zero, to avoid large deformations that would lead to a degenerate solution. They are defined as an $l_2$-norm loss:

$$\mathcal{L}_{\text{rot}}^D = \|\mathbf{e}\|_2, \tag{9}$$
$$\mathcal{L}_{\delta \mathbf{v}}^D = \|\delta \mathbf{v}\|_2. \tag{10}$$

**Position consistency loss $\mathcal{L}_{pos}^D$:** Following (Xie et al., 2024), we introduce a consistency prior on the deformation of neighboring triangles. The underlying intuition is that spatially close triangles in the canonical space should remain similarly close after deformation. This loss encourages the relative distances between the $K$ nearest neighbors to be preserved after deformation, thus promoting coherent local movements and reducing the risk of degenerate distortions. The position consistency loss is formally defined as:

$$\mathcal{L}_{pos}^D = \sum_{n=0}^{N_T} \sum_{k=1}^{K} \left\| d\left(\mathbf{x}_c^{(n)}, \mathbf{x}_c^{(k)}\right) - d\left(\mathbf{x}_o^{(n)}, \mathbf{x}_o^{(k)}\right) \right\|_1, \tag{11}$$

where $\mathbf{x}_c^{(n)}$ and $\mathbf{x}_o^{(n)}$ denote the center of the $n$-th triangle in canonical and deformed space, respectively, and $d(\cdot, \cdot)$ represents a Euclidean distance.

## 3. Experimental results

### 3.1. Implementation details

We initialize the scene following Surgical Gaussian (Xie et al., 2024), without applying masks to remove the surgical tools. Canonical triangles are initialized with the same parameters as triangle splatting (Held et al., 2026), and for the first 100 iterations only the canonical representation is optimized. Then the deformation network is initialized and optimized up to 40k iterations. Densification and pruning are done every 500 iterations and stop after 5k iterations, and the canonical scene is frozen after 6k iterations, as further optimization provides no benefit.

We present our experimental results on the Endo-NeRF (Wang et al., 2022) dataset, which consists of two robotic prostatectomy stereo videos recorded with a static camera, with different deformations, such as *pulling* and *cutting* of soft tissues. Since our method is designed for monocular inputs, we ignore all stereo information and only use the left frames for both input and depth estimation. Additionally, the video is processed in smaller sequences to avoid the instability introduced by long-range temporal dependencies. This reduces optimization complexity, confines the deformation network to local motion, and

yields more stable and efficient convergence. For quantitative evaluation, we provide a Peak Signal-to-Noise Ratio (PSNR) (Horé and Ziou, 2010), Structural Similarity Index Measure (SSIM) (Wang et al., 2004) and Learned Perceptual Image Patch Similarity (LPIPS) (Zhang et al., 2018).

For further experimental validation, we considered the dataset introduced in NeRFscopy (Salort-Benejam and Agudo, 2026), which is composed of four in-vivo monocular surgical videos: two Totally Endoscopic Coronary Artery Bypass (TECAB) procedures (Agudo, 2021; Stoyanov et al., 2005), a lung lobectomy (Giannarou et al., 2013), and a bronchoscopy (Urdapilleta and Agudo, 2023). All these sequences exhibit mild to severe tissue deformations, different anatomies, and varying illumination conditions, which makes them more challenging.

Additionally, we also used a clip from the StereoMIS (Hayoz et al., 2023) dataset, captured during Da Vinci robotic surgery in in-vivo porcine subjects. For qualitative and quantitative evaluation of this scene see Appendix A.

### 3.2. Loss terms ablation

We conducted an ablation analysis on the main loss components of our model using the *pulling* sequence. Starting from the full Endo-4DTS loss in Equation (5), we removed each term individually: the normal deformation loss $\mathcal{L}_{\mathbf{N}}^{D}$, the normal canonical loss $\mathcal{L}_{\mathbf{N}}^{C}$, the positional consistency loss $\mathcal{L}_{pos}^{D}$, the depth smoothness loss $\mathcal{L}_{smooth}^{C}$ and the depth loss $\mathcal{L}_{depth}^{C}$. Quantitative results in Table 1 show an unexpected behavior in which omitting all new losses yields the best photometric results. These metrics, however, do not capture geometric consistency. Qualitative inspection, see Figure 2, reveals that removing all losses produces visually sharp renderings but significantly noisier and less consistent depth and normal estimations, indicating that the deformation network becomes under-constrained and overfits to the photometric cues. Adding the depth loss leads to the largest improvement, removing depth discontinuities and stabilizing normals, though specular-related artifacts persist. Incorporating the normal losses mitigates these effects and improves rendering and normal quality, although minor depth floaters remain.

Table 1: **Quantitative results of the ablation analysis** on the *pulling* sequence. Best results in **bold**, second best underlined.

|  | PSNR↑ | SSIM ↑ | LPIPS ↓ |
|---|---|---|---|
| Endo-4DTS | 38.113 | 0.958 | 0.044 |
| w/o $\mathcal{L}_{\mathbf{N}}^{D}$ | 37.667 | 0.956 | 0.046 |
| w/o $\mathcal{L}_{\mathbf{N}}^{C}$ | 37.116 | 0.953 | 0.046 |
| w/o $\mathcal{L}_{pos}^{D}$ | 37.661 | 0.955 | 0.045 |
| w/o $\mathcal{L}_{smooth}^{C}$ | 37.411 | 0.953 | 0.047 |
| w/o $\mathcal{L}_{depth}^{C}$ | **38.304** | **0.959** | **0.043** |

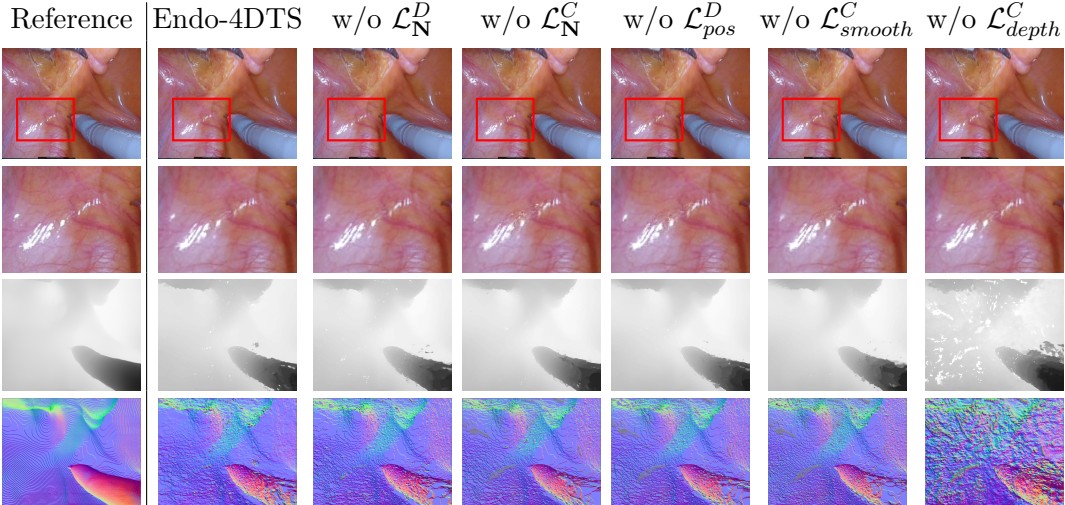

Figure 2: **Qualitative results of the ablation analysis** on a frame from the *pulling* sequence. **Row 1:** RGB images. **Row 2:** Zoomed in RGB images. **Row 3:** Surface depth. **Row 4:** Surface normal.

### 3.3. Sensitivity study

We conducted a sensitivity analysis of the loss weigths $\lambda_{11}$ and $\lambda_{12}$ associated with the rotation and translation regularizations, motivated by the fact that removing these terms caused the deformation network to become too under-constrained. As shown in Table 2, reducing either weight consistently degrades performance, indicating that stronger regularization improves optimization stability. Qualitatively, see Figure 3, this is confirmed by enhanced rendering quality and more visually accurate geometry.

Table 2: **Quantitative results of the sensitivity study** on $\lambda_{11-12}$ on the *pulling* sequence. Best results in **bold**.

| $\lambda$ values | PSNR↑ | SSIM ↑ | LPIPS ↓ |
|---|---|---|---|
| 0.5 | **37.667** | **0.956** | **0.046** |
| 0.1 | 36.499 | 0.951 | 0.054 |
| 0.01 | 35.654 | 0.946 | 0.0632 |

### 3.4. Final results

**EndoNeRF dataset.** We provide a quantitative and qualitative assessment of the performance of final version of our Endo-4DTS. We compare our results with other methods that apply NeRF-based or Gaussian-based approaches to the Endo-NeRF (Wang et al., 2022) dataset in Table 3. We cannot directly compare our results with any other method

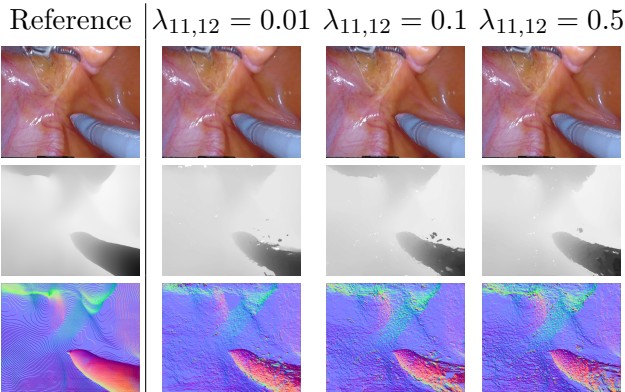

Reference | $\lambda_{11,12} = 0.01$ | $\lambda_{11,12} = 0.1$ | $\lambda_{11,12} = 0.5$

Figure 3: **Qualitative results of the sensitivity study** on $\lambda_{11-12}$ on the *pulling* sequence. **Row 1:** RGB images. **Row 2:** Surface depth. **Row 3:** Surface normal.

using triangle splatting (Held et al., 2026) as we are the first, to our knowledge, to extend this recent work to dynamic scenes.

Table 3: **Quantitative comparison** of our method Endo-4DTS with EndoNeRF (Wang et al., 2022), EndoSurf (Zha et al., 2023), LerPlane (Yang et al., 2023a), Endo-4DGS (Huang et al., 2024), EndoGaussian (Liu et al., 2024), and SurgicalGaussian (Xie et al., 2024) on the EndoNeRF (Wang et al., 2022) dataset. Best results are highlighted in **bold**.

| Methods | "pulling" | | | "cutting" | | |
|---|---|---|---|---|---|---|
| | PSNR↑ | SSIM ↑ | LPIPS ↓ | PSNR ↑ | SSIM ↑ | LPIPS ↓ |
| EndoNeRF | 34.217 | 0.938 | 0.160 | 34.186 | 0.932 | 0.151 |
| EndoSurf | 35.004 | 0.956 | 0.120 | 34.981 | 0.953 | 0.106 |
| LerPlane | 36.241 | 0.950 | 0.102 | 35.580 | 0.955 | 0.101 |
| Endo-4DGS | 36.56 | 0.955 | 0.032 | 37.85 | 0.959 | 0.043 |
| EndoGaussian | 37.308 | 0.958 | 0.070 | 38.287 | 0.962 | 0.058 |
| SurgicalGaussian | 38.783 | 0.970 | 0.049 | 37.505 | 0.961 | 0.062 |
| Endo-4DTS (Ours) | **40.39** | **0.971** | **0.026** | **38.876** | **0.966** | **0.029** |

As can be seen, our method consistently outperforms all other approaches, demonstrating its superior representation capacity and rendering quality. The improvement is particularly pronounced in the LPIPS metric, which highlights the ability of our Endo-4DTS to generate more visually realistic images. The qualitative results in Figure 4 and Figure 5 further confirm these findings, capturing details like tissue capillaries and specular highlights with high-fidelity, without compromising the geometric properties of the scene.

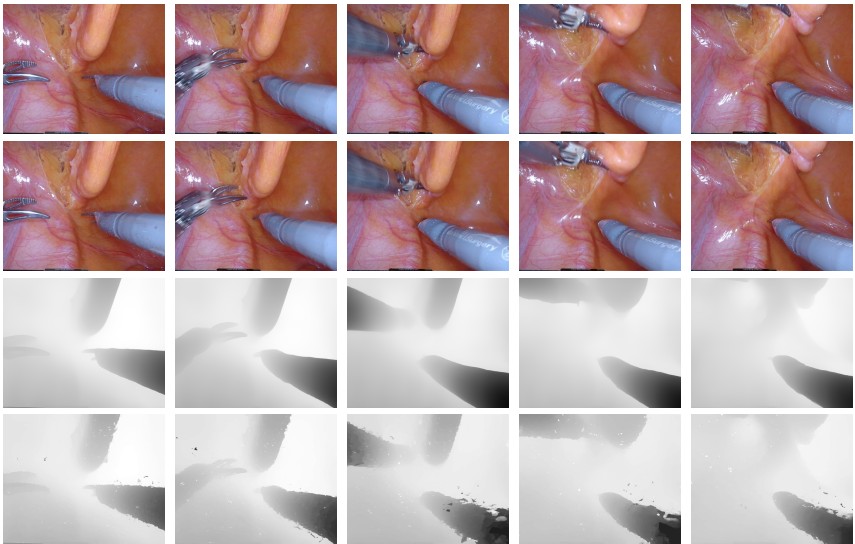

Figure 4: **Rendered results of Endo-4DTS model on five frames of the *pulling* video. Row 1:** RGB input. **Row 2:** RGB output. **Row 3:** Depth input. **Row 4:** Depth output.

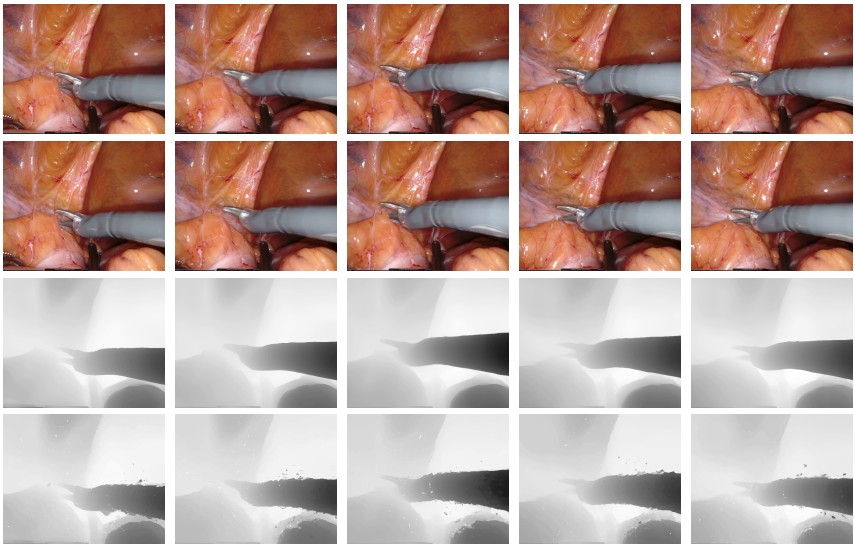

Figure 5: **Rendered results of Endo-4DTS model on five frames of the *cutting* video. Row 1:** RGB input. **Row 2:** RGB output. **Row 3:** Depth input. **Row 4:** Depth output.

However, while the estimated photometric properties of the scene are remarkable, the surface depth estimation presents small localized inconsistencies, with a small number of triangles that are not perfectly aligned with the surrounding surface. These triangles are typically displaced only slightly in front of or behind the main surface, rather than appearing at extreme depths. Such errors in the estimation of the geometric properties may arise because the MLP tends to prioritize photometric accuracy over the depth regularization and other geometry-related objectives during optimization. Additionally, we observe a larger number of misplaced triangles around the moving surgical tools, which occlude the tissues behind them and make it more difficult for the MLP to learn where to correctly place those triangles when a new part of the tissue–at a considerable different depth than the tool–is revealed. Moreover, the densification strategy could introduce some undesired noise to the triangles, given that it was taken directly from triangle splatting (Held et al., 2026) which only considers rigid scenes. The design of densification and pruning strategies specifically tailored for deformable scenes is an interesting and promising line for future research, which we plan to explore. In the future, we will also consider more sophisticated geometric priors that constrain the motion of triangles, in order to simultaneously guarantee both geometric and photometric properties.

**NeRFscopy dataset.** To evaluate the generalization capabilities of our proposed method, we additionally tested our method on the NeRFscopy dataset (Salort-Benejam and Agudo, 2026). Table 4 presents a quantitative comparison between Endo-4DTS, NeRFscopy (Salort-Benejam and Agudo, 2026) and EndoNeRF (Wang et al., 2022). Again Endo-4DTS consistently outperforms previous ones across all metrics by relevant margins. Qualitative results are shown in Figure 6, where our method produces accurate RGB renderings and plausible geometrical properties across different scenarios.

Our method requires on average 3 to 5 hours to train on a single GPU NVIDIA RTX A5000. Training and rendering times strongly depend on both the number of triangles used to represent the scene and the resolution of the rendered images; in our experiments, we observe an average rendering speed of 17.74 FPS for the EndoNeRF (Wang et al., 2022) dataset, 26.86 FPS for the NeRFscopy (Salort-Benejam and Agudo, 2026) dataset and 16.84 FPS for the StereoMIS (Hayoz et al., 2023) dataset. We emphasize that the primary focus of this work was demonstrating the feasibility and effectiveness of extending triangle splatting (Held et al., 2026) to deformable endoscopy environments, rather than optimizing computational efficiency which will be addressed in future work. In any case, Endo-4DTS obtains a remarkable tradeoff between accuracy and computational efficiency.

## 4. Conclusions

We introduced Endo-4DTS, a self-supervised method for synthesis of deformable endoscopy scenes by extending triangle splatting (Held et al., 2026) to non-rigid environments through an explicit canonical representation of the scene that is jointly optimized with a deformation network modeling the triangle transformations. Our sophisticated loss design stabilizes training and enables learning an explicit, deformable representation of the scene. Experimental results demonstrate state-of-the-art performance across PSNR, SSIM, and LPIPS, with consistently sharper and coherent renderings, highlighting the superiority of our method. To the best of our knowledge, we are the first to successfully extend trian-

gle splatting (Held et al., 2026) to deformable environments, and the first to apply it to endoscopy scenes.

Table 4: **Additional experiments** of our method Endo-4DTS, compared with EndoN-eRF (Wang et al., 2022) and NeRFscopy (Salort-Benejam and Agudo, 2026) on the NeRFscopy (Salort-Benejam and Agudo, 2026) dataset. Best results are highlighted in **bold**.

|  |  | PSNR ↑ | SSIM ↑ | LPIPS↓ |
|---|---|---|---|---|
| | EndoNeRF | 25.791 | 0.742 | 0.255 |
| TECAB1 | NeRFscopy | 25.811 | 0.750 | 0.255 |
| | Endo-4DTS | **28.26** | **0.863** | **0.136** |
| | EndoNeRF | 24.954 | 0.685 | 0.419 |
| TECAB2 | NeRFscopy | 24.864 | 0.689 | 0.429 |
| | Endo-4DTS | **25.983** | **0.783** | **0.238** |
| | EndoNeRF | 27.142 | 0.788 | 0.293 |
| Lung Lobectomy | NeRFscopy | 27.285 | 0.791 | 0.275 |
| | Endo-4DTS | **28.435** | **0.825** | **0.178** |
| | EndoNeRF | 33.872 | 0.867 | 0.588 |
| Bronchoscopy | NeRFscopy | 34.405 | 0.875 | 0.156 |
| | Endo-4DTS | **35.566** | **0.902** | **0.080** |

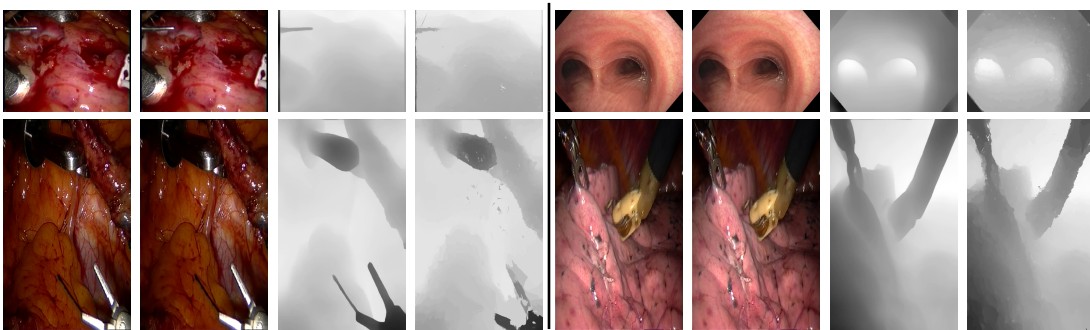

Figure 6: **Qualitative evaluation on the NeRFscopy dataset.** In both sides, the same information is displayed. Columns show from left to right: arbitrary input frame, rendered RGB output, input depth estimation, and rendered depth output. **Left:** TECAB1 and TECAB2 images. **Right:** Bronchoscopy and lung lobectomy images.

## Acknowledgments

This work has been supported by the project GRAVATAR PID2023-151184OB-I00 funded by MCIU/AEI/10.13039/501100011033 and by ERDF, UE; and by the Government of Catalonia under 2025 FI-STEP 00398.

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

## Appendix A. Supplementary experiments

To further evaluate the performance of our Endo-4DTS method, we also used one sequence from the StereoMIS (Hayoz et al., 2023), specifically, the first 110 frames of sequence "P2_7" as done in SurgicalGS (Chen et al., 2025a). Table 5 reports a quantitative comparison between several NeRF- and Gaussian-based approaches. As shown, Endo-4DTS consistently outperforms all previous state-of-the-art methods across all evaluated metrics.

These quantitative results are supported by the qualitative results presented in Figure 7, where several frames of the sequence can be seen. Our method produces renderings with high photometric fidelity, accurately representing fine details and texture. Moreover, the estimated depth maps are temporally and geometrically coherent on the tissue surfaces. However, the depth around the surgical instruments exhibits more artifacts and worse geometrical quality.

Table 5: **Quantitative comparison** of our method Endo-4DTS with EndoNeRF (Wang et al., 2022), EndoSurf (Zha et al., 2023), LerPlane (Yang et al., 2023a), EndoGaussian (Liu et al., 2024), Deform3DGS(Yang et al., 2024b) and SurgicalGS (Chen et al., 2025a) on the StereoMIS (Hayoz et al., 2023) dataset. Best results are highlighted in **bold**.

| Methods | PSNR↑ | SSIM ↑ | LPIPS ↓ |
|---|---|---|---|
| EndoNeRF | 28.79 | 0.809 | 0.266 |
| EndoSurf | 29.36 | 0.861 | 0.211 |
| LerPlane | 29.09 | 0.789 | 0.179 |
| EndoGaussian | 31.02 | 0.878 | 0.132 |
| Deform3DGS | 31.61 | 0.888 | 0.135 |
| SurgicalGS | 31.54 | 0.885 | 0.148 |
| Endo-4DTS (Ours) | **32.110** | **0.903** | **0.076** |

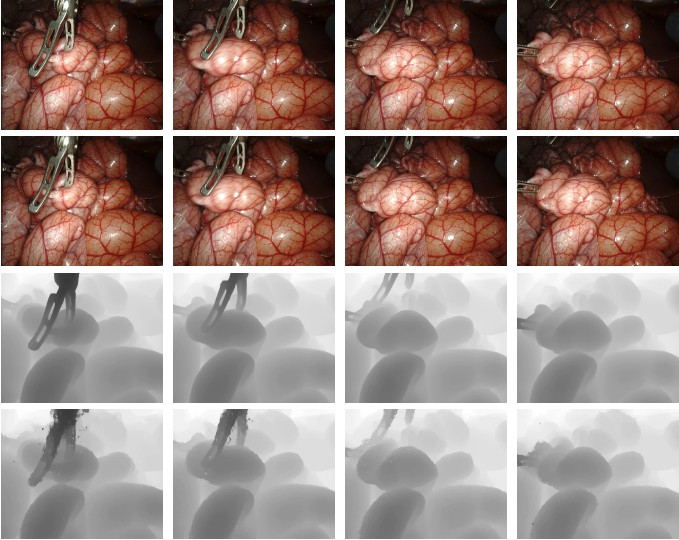

Figure 7: **Rendered results of Endo-4DTS model on four frames of the StereMIS (Hayoz et al., 2023) video. Row 1:** RGB input. **Row 2:** RGB output. **Row 3:** Depth input. **Row 4:** Depth output.

