# OpenReview forum: "Endo-4DTS: Monocular 4D Scene Synthesis for Endoscopy via Deformable Triangle Splatting"
_MIDL.io/2026/Conference — MIDL 2026 Poster_

### Official Review · Reviewer_KcMK · 2025-12-29

**Confidence:** 3
**Preliminary Rating:** 4
**Final Rating:** 4

**Summary:**

This paper introduces Endo-4DTS, a SSL framework for monocular 4D scene synthesis in deformable endoscope videos. The authors extends triangle splatting to deformable setting and jointly optimize triangles with a deformable network that predicts time-depenedent transformations. The authros take into account of geometry (e.g., depth, normal, smoothness) in their final loss. The authors demonstrated superior performance compared to NeRF and Gaussian splatting absed baselines, especially on the LPIPS metric.

**Strengths:**

- This paper is a meaningful adaptation of triangle splatting to the medical domain, e.g., time-varying, non-rigid setting.
- Consistent performance improvement compared to baselines, though the performance can be marginal for PSNR/SSIM metrics but showed significant gain in LPIPS metrics.

**Weaknesses:**

- Ablation in Table 1 reveals that removing some losses improve photometric metrics but lead to less consistent depth/normal estimation. This suggest that the authors should prioritize include geometry related metrics to better characterize model performance.
- Assumption of fixed camera pose across time seems to be unrealistic for endoscopic videos. Some discussion and characterization of this assumption would be helpful.

**Detailed Comments:**

n/a

**Justification Of Final Rating:**

The authors sufficiently addressed my concerns. In particular, they mentioned how it's challenging to compute geometric metrics due to lack of ground-truth. This is reasonable. I'll keep my rating of weak accept.

**Justification Of The Preliminary Rating:**

The paper presents a technically sound and novel extension of triangle splatting to deformable endoscopic scenes, with clear empirical gains that are particularly in perceptual quality (LPIPS). While some assumptions (e.g., static camera) and limited geometric evaluation weaken the overall impact, the method is well-motivated, carefully implemented, and shows strong potential for medical 4D reconstruction, justifying a weak accept.

**Questions To Address In The Rebuttal:**

see weaknesses

---

> ### Author Response · Authors · 2026-01-25
>
> Static camera. We acknowledge that camera motion is common in some endoscopic procedures, such as gastroscopies or exploratory surgery. However, having a static camera is a reasonable assumption for many endoscopy settings, where the endoscope is intentionally kept static to maintain a stable field of view and allow precise manipulation of the tissues. In this context, while the camera is fixed the tissues can deform along time. We have clarified this point in the revised paper, page 4.
>
> Geometry related metrics. Unfortunately, when working with these type of sequences we often lack ground-truth geometric information. During endoscopy procedures, the camera used can be monocular--which is our case-- or stereo, but no other equipment can be used to recover reliable 3D information that would allow direct geometry evaluation.

---

### Official Review · Reviewer_aDQC · 2026-01-07

**Confidence:** 2
**Preliminary Rating:** 4
**Final Rating:** 5

**Summary:**

This paper presents Endo-4DTS, a novel self-supervised framework capable of synthesizing 4D deformable endoscopic scenes from monocular videos by extending triangle splatting to non-rigid environments. The method utilizes a canonical set of triangles jointly optimized with a deformation network and incorporates geometric priors from monocular depth estimation to ensure structural consistency across time. Extensive experiments indicate that Endo-4DTS significantly outperforms existing state-of-the-art approaches in terms of rendering quality and geometric accuracy on standard endoscopic datasets.

**Strengths:**

1. The paper presents a novel adaptation of triangle splatting for deformable scenes, marking the first application of this efficient rendering primitive to the challenging domain of monocular 4D endoscopy.

2. By integrating depth and normal priors with a position consistency loss, the method effectively constrains the deformation network to produce geometrically coherent reconstructions despite the ill-posed nature of monocular non-rigid estimation.

3. Experimental results demonstrate that the proposed framework consistently achieves superior visual fidelity and higher quantitative metrics compared to existing state-of-the-art NeRF and Gaussian Splatting approaches.

**Weaknesses:**

1. The assumption of a static camera pose limits the method's practical applicability in real-world endoscopic procedures where continuous and complex camera motion is the norm.
2. The densification and pruning strategies are directly adopted from rigid scene formulations, which leads to geometric artifacts such as floating triangles and depth discontinuities in deformable regions.
3. The framework relies heavily on pseudo-ground truth depth maps from a pre-trained external model, implying that the geometric accuracy is inherently upper-bounded by the errors of this auxiliary estimator.
4. The optimization process tends to prioritize photometric fidelity over geometric regularization, causing the model to occasionally produce inconsistent surface normals and noisy mesh reconstructions.
5. Experimental validation is currently limited to the Endo-NeRF dataset, resulting in a lack of comprehensive evidence regarding the method's generalization capabilities across diverse surgical interventions and anatomical structures.

**Detailed Comments:**

Please refer to the weaknesses.

**Justification Of Final Rating:**

The authors have thoroughly addressed my concerns, particularly regarding the experimental scope, by validating the method on two additional datasets (NERFscopy and StereoMIS). Given the novel application of triangle splatting to deformable tissues and the demonstration of superior performance and generalization, I am increasing my rating to Strong Accept.

**Justification Of The Preliminary Rating:**

While the current limitation to static cameras restricts immediate clinical utility, the paper proposes a novel and effective extension of triangle splatting to deformable environments that achieves state-of-the-art rendering quality.

**Questions To Address In The Rebuttal:**

Please refer to the weaknesses.

---

> ### Author Response · Authors · 2026-01-25
>
> Static camera. We acknowledge that camera motion is common in some endoscopic procedures, such as gastroscopies or exploratory surgery. However, having a static camera is a reasonable assumption for many endoscopy settings, where the endoscope is intentionally kept static to maintain a stable field of view and allow precise manipulation of the tissues. In this context, while the camera is fixed the tissues can deform along time. We have clarified this point in the revised paper, page 4.
>
> Densification and pruning. We agree with the reviewer that our current densification and pruning strategies are not tailored for deformable scenes, which could potentially introduce discontinuities in certain regions. However, our work was primarily focused on demonstrating the feasibility of extending Triangle Splatting for deformable endoscopy scene synthesis. We believe that incorporating deformation-aware densification and pruning strategies could improve the geometric representation of the scene and is a promising direction for future work, which is now reflected in the revised paper, in page 11.
>
> Pre-trained depth estimation model and optimization process. We agree with the reviewer that our framework relies on pseudo-ground truth depth for regularization, as there is no depth information available when using monocular videos, we can only rely on photometric information for supervision. However, the depth loss in Eq. (6) is only applied to the canonical space, to provide a stable starting point for the deformation network, which remains free to learn physically plausible and temporally consistent deformations that better fit the observed data. Moreover, the normal loss in Eq. (8) is applied to the deformed space, acting as a soft geometric regularizer rather than a hard constraint. We assume that with this architecture our model is not strictly upper-bounded by the pseudo-ground truth depth input.
>
> Experimental validation. Due to space constraints, we proposed our experimental evaluation on the EndoNeRF dataset, the most commonly used dataset in this context, providing a full comparison with respect to state-of-the-art methods as well as both quantitative and qualitative analysis. We now extend our validation by considering two additional datasets: 1) NERFscopy composed of four videos with very different procedures, geometries and illumination conditions, and 2) StereoMIS captured during Da Vinci robotic surgery in in-vivo procine subjects. This information is included at the end of section 3.4 and in page 7. Our results are reported and discussed in Table 4 and Figure 6 in page 12, and in Appendix A, Table 5 and Figure 7. Thanks to that, we can highlight the generalization capabilities of our method and its superiority with respect to previous ones, showing exhaustive experimental evaluation in a wide variety of real and challenging videos.

---

### Official Review · Reviewer_evR3 · 2026-01-08

**Confidence:** 3
**Preliminary Rating:** 4
**Final Rating:** 4

**Summary:**

The paper focuses on applying triangle splatting to the domain of endoscopy images. The novelty of the paper revolves on extending the original triangle splatting work to deformable scenes, a requirement for endoscopy domains due to the nature of soft tissues and the movement of objects in the camera’s views. The method outperforms existing methods based on NeRF/Gaussians.

**Strengths:**

* The paper offers an overview of concurrent scene-modelling approaches for endoscopy data, providing a clear outline of their limitations, and establishes a solid foundation on which to ground their contributions.
* The proposed work combines well-established approaches to build their architecture. Additionally, the method uses endoscopy-specific pseudo-groundtruths to further constrain their method.
* On top of out-performing existing approaches, the efficiency of triangle splatting has strong implications for the real-time usability in clinical practice. The ablation of the training objective is thoroughly investigated.

**Weaknesses:**

* The optimization and rendering times of the approach is lacking in the paper. One key benefit of triangle splatting alongside its simplicity, is the suitability of its operations for GPU acceleration. Readers interested in clinical application would certainly appreciate to see these results, as well as the times for the NeRF/GaussianSplat baselines, as this is one of the main selling points of the original triangle splat paper.

* The method lacks a reference to Nerfies (Park. 2021), arguably one of the most notable papers in the deformable scene domain.

* The effect of the design of the deformation field is hardly explored. While INRs for deformation modelling has been thoroughly explored, little is discussed in the manuscript. The floating triangles in the final depth maps could be a result of the lack of smoothness in the deformation field, but this is not discussed. The overall quality of the deformation field is not shown nor discussed. While the reconstruction results look promising, there is a surprising amount of discontinuities/floaters in the qualitative results of the depth and normal maps whose root cause is simply brushed over.

**Detailed Comments:**

* Some additional clarity would be helpful in the triangle splatting section of the method. The ranges of parameters would be helpful for new audiences looking to understand triangle splatting. While this is trivial for the opacities ([0,1]), the range of the sigma parameter could use further clarification (the reader can be pointed to the original work for further clarification).

* As already mentioned, optimization and rendering times are missing from the manuscript. This would be helpful to understand how close the work brings this technology to clinical usability. While there is little space left in the manuscript, I’d argue those details are more important over eg. the amount of qualitative results displayed.

* A paper putting forward a novel approach such as this one, should have an unbiased discussion of the limitations and issues. As an example, Nerfies 2021 has well discussed limitations and failure cases. This paper does not offer this. I’d much rather have some of the ablation be an appendix section and see a more honest discussion of the results.

**Justification Of Final Rating:**

I want to thank the authors for their time and effort put into the rebuttal. The added performance metrics were helpful and confirm the clinical utility. The discussion regarding deformation modeling clarified the approach while it also highlights some limitations, e.g. the lack of spatial continuity.

**Justification Of The Preliminary Rating:**

The idea is novel, its performance for endoscopy data is promising, and it displays important technical benefits for clinical application. However, there are performance details left undiscussed. Concretely, optimization/rendering speeds are not presented and limitations in the form of failure cases are not presented.

**Questions To Address In The Rebuttal:**

* The manuscript argues there are discontinuities and ‘floating’ triangles far way from the input depth map. However, Figures 4 and 5 appear to show some of these discontinuities are depths similar to the objects they are close to (not directly in front of the camera or far away behind the scene). Are these fully transparent? If not, how is there so much variation in the position of these floating triangles over time? Where do these triangles go over the course of the deformation? Is the deformation field in those regions turbulent enough to allow floating triangles to be ‘moved out of the surface’ of the objects? Or are those triangles coming from outside the camera’s view?

* There is little-to-no results showing the quality of the deformation fields.
* Why is there no strict regularization of the deformation field? While the authors present a distance-preserving neighbourhood loss, this may not be enough to preserve local structure. Are there any guarantees that the space does not locally fold?

Currently the paper offers too little discussion on the source of these shortcomings. This lack of transparency may stop future readers from being able to correctly determine if this method applicable to their down-stream tasks.

---

> ### Author Response · Authors · 2026-01-25
>
> Optimization and rendering times. We thank the reviewer for highlighting the importance of disclosing optimization and rendering times for clinical applicability. In the current work, our primary focus was demonstrating the feasibility and effectiveness of extending Triangle Splatting to deformable endoscopy environments, rather than optimizing computational efficiency. For reference, our method takes on average 3 to 5 hours to train on a single GPU NVIDIA RTX A5000. Afterwards, rendering times strongly depend on the number of triangles used to represent the scene and the size of the rendered images, in our experiments we observe an average rendering speed of 17.74 FPS for the EndoNeRF dataset, 26.86 FPS for the NeRFscopy dataset and 16.84 FPS for the StereoMIS dataset. These numbers are promising for this type of clinical intervention.
> We have added a brief discussion of the optimization and rendering times in the revised manuscript, and clarified that computational efficiency is not the primary focus of this study, see page 11.
>
> Nerfies reference. We thank the reviewer for highlighting this relevant work. We have included and discussed some NeRF-based works to handle non-rigid bodies. This is considered in the revised paper, page 2.
>
> Deformation field quality. Regarding the deformation field, we would like to point out that our method, like any splatting-based one, is depicted as an explicit scene representation instead of an implicit one. This type of methods are quite different from INRs as the deformation network parametrizes deformations of explicit primitives, not a continuous field in space. The smoothness properties often evaluated in NeRF-based approaches might not directly apply to our deformation network. For this reason, we incorporate several losses to impose local smoothness in the scene, such as the pose consistency loss (see Eq.(11)) and the depth smoothness one (see Eq.(7)).
>
> Ranges of parameters in TS. We thank the reviewer for this suggestion. The smoothness parameter, $\sigma$, can only take positive values, and it controls the sharpness of the window function in Eq. (2), being a trainable parameter as the opacity. This has been clarified in the revised paper, page 3.
>
> Discussion of limitations. We thank the reviewer for pointing out that our discussion was too short. We have tried to explain some more possible sources of error in the geometry estimation in page 11.
>
> Geometry discontinuities. We would like to apologize to the reviewer for any confusion caused by our imprecise wording. By "discontinuities and floating triangles far away from the input depth map" we did not mean that the triangles were located at extreme depths. Rather, we observed that a small number of triangles in the rendered depth maps were misaligned with the surrounding surface. In particular, around the surgical tools, some triangles appear slightly in front of or behind the main surface, rather than being coherently aligned with it. We have rephrased the sentence in page 11 in the revised paper to clarify this and avoid any further misunderstandings.
>
> Deformation field regularization. We thank the reviewer for raising this concern and apologize for the confusion. We would like to point out that the pose consistency loss is not the only term in our global loss that is directly regularizing the deformation field, as it can be seen in Eq.(5) of the paper. Particularly, all the terms with the superscript \textbf{$^D$} help to this task. Note that the rotation and translation loss terms (see Eqs.(9)-(10)) also regularize the deformation network to avoid large deformations. Additionally, we incorporate a normal loss (see Eq.(8)) for the deformed space which also regularizes the deformation field to ensure the estimated normals in the deformed space align with the pseudo-ground truth surface normals from the input depth.

---

> > ### Comment · Reviewer_evR3 · 2026-01-29
> >
> > I would like to thank the authors for the thoughtful rebuttal. I would like to continue the discussion regarding the deformation modeling, acknowledging that we are in the discussion phase and I am not expecting new experiments, ablations, or changes to the revised manuscript.
> >
> > *On optimization and rendering times:*
> > I am happy to have the authors share the optimization and inference times with us. I share the author's remarks regarding the potential for clinical adoption and apreciate the updated discussion, as I believe this is a strong selling point of your work.
> >
> > *On deformation modeling:*
> >  Thank you for the clarification on the deformation modeling. I do see that you are indeed modeling the deformation of individual triangles, and not the space in which they live. Could you provide some clarification on why you chose to model it this way? Is this standard practice in explicit literature? Being familiar with deformation modeling approaches, I find it surprising that this approach would be prefered over modeling the deformations as a vector field. Modeling the deformation as a vector field has well known techniques from eg. the image registration literature. By, for example,  enforcing the space to not fold (via Jacobian determinant or bending energy losses), you could mitigate the issue with triangles 'escaping' the surface. In your current implementation, it appears you have to resort to weaker regularization losses that minize magnitudes (Eq. 9-10) and changes in relative positions (Eq. 11). Are there works that describe the benefits/pitfalls of this triangle-wise approach? Or alternatively, experiences of your own that drove you to model it in such a way?

---

> > > ### Author Response · Authors · 2026-01-30
> > >
> > > We thank the reviewer for initiating this discussion and for helping us improve our work. It is true that the most natural way to model deformations is through a deformation field, especially in simulation and image registration based on physical models, such as those using finite elements. However, in artificial vision, this approach is not usually the most natural, since each point requires three degrees of freedom, and only two (in the geometric case) or one (in the photometric case) equations are available to learn those values. Consequently, for example, in non-rigid structure from motion, the use of probabilistic models based on subspaces (a basis in shapes, trajectories, or forces could be used here) has been widely adopted in the last decade. These models allow us to address a problem that is inherently ill-conditioned. Moreover, this formulation has been widely adopted in explicit 3D reconstruction methods based on gaussian splatting for deformable environments and proved to be efficient for recovering the scene geometric and photometric properties. For these reasons, we decided to implement this type of modeling to extend the triangle splatting approach for deformable endoscopic environments. That said, we use triangles due to their great versatility, as they are the simplest possible primitive for modeling deformation (as is the case in simulation, where triangles are also widely used). By learning only six parameters (three angles and three translations) we are able to adapt the entire triangular geometry, ensuring consistency with the information observed in the image. In other words, we have a piecewise model that can adapt to a wide variety of geometries. Since a mesh effect has not been imposed, we include additional regularizers, which allow for better geometry adjustments and improved conditioning. This spatial continuity is something we want to explore further in future work. Finally, we would like to emphasize that current GPU-based hardware is ultra-efficient for processing and rendering triangles, making it a key path to achieving an accurate and efficient solution.

---

### Author Rebuttal · Authors · 2026-01-25

**Rebuttal:**

We thank the reviewers for their valuable feedback and are pleased they found our work establishes a solid foundation on which to ground our contributions (evR3); our work represents a novel and significant self-supervised framework capable of synthesizing 4D deformable endoscopic scenes from monocular videos marking the first application of this efficient rendering primitive to a challenging domain (aDQC,KcMK); the efficiency of triangle splatting has strong implications for the real-time usability in clinical practice even out-performing existing approaches (evR3) by means of extensive experiments (aDQC,KcMK). Next, we address all the concerns in the respective boxes. We have also uploaded a revised paper, highlighting all changes in red.

**Supporting Material:**

/attachment/2d5fab12d835f11adb88655617b316ef1b6f3c13.zip

---

### Meta-Review · Area_Chair_dj5D · 2026-02-09

**Recommendation:** Accept (Poster)
**Confidence:** 4

**Metareview:**

Initial review already skewed positive and after revisions, all reviewers are in agreement for acceptance.

---

### Decision · Program_Chairs · 2026-02-13

Accept (Poster)